# GreedyFool: Distortion-Aware Sparse Adversarial Attack

**Xiaoyi Dong**[1][*] **Dongdong Chen**[2][†] **Jianmin Bao**[2], **Chuan Qin**[1],
**Lu Yuan**[2], **Weiming Zhang**[1], **Nenghai Yu**[1], **Dong Chen**[2]
[1]University of Science and Technology of China [2]Microsoft Research
`{dlight@mail., qc94@mail., zhangwm@, ynh@ }.ustc.edu.cn`
`cddlyf@gmail.com , {jianbao, luyuan, doch }@microsoft.com`

## Abstract

Modern deep neural networks(DNNs) are vulnerable to adversarial samples. Sparse adversarial samples are a special branch of adversarial samples that can fool the target model by only perturbing a few pixels. The existence of the sparse adversarial attack points out that DNNs are much more vulnerable than people believed, which is also a new aspect for analyzing DNNs. However, current sparse adversarial attack methods still have some shortcomings on both sparsity and invisibility. In this paper, we propose a novel two-stage distortion-aware greedy-based method dubbed as "GreedyFool". Specifically, it first selects the most effective candidate positions to modify by considering both the gradient(for adversary) and the distortion map(for invisibility), then drops some less important points in the reduce stage. Experiments demonstrate that compared with the start-of-the-art method, we only need to modify $3\times$ fewer pixels under the same sparse perturbation setting. For target attack, the success rate of our method is 9.96% higher than the start-of-the-art method under the same pixel budget. Code can be found at https://github.com/LightDXY/GreedyFool.

## 1 Introduction

Despite the success of current deep neural networks (DNNs), they are shown to be vulnerable to adversarial samples [41, 17, 6, 36, 24, 31, 4, 18, 42, 13, 14, 47]. An adversarial sample is a carefully crafted image to fool the target network by adding small perturbations onto the original clean image. On one hand, the existence of adversarial samples raises the security threat of DNNs, especially these DNNs used in current daily life [23]. On the other hand, adversarial samples can help us to disclose the weakness of DNNs, and further understand the mechanism of DNNs.

To understand the weakness and mechanism of DNNs comprehensively, it is necessary to achieve various kinds of adversarial attacks. In most cases, adversarial perturbations are constrained by a $l_p$-norm distance and can be roughly categorized into two types: dense attack and sparse attack. For dense attack, the attack methods perturb all the pixels of the image under the $l_\infty$ or $l_2$ constrain [10, 6, 41]. For sparse attack, the attack methods only perturb a few pixels under the $l_0$ constrain [34, 30, 11].

Since the pioneering work [41], many dense attack methods have been proposed and achieved impressive results [15, 35, 5, 10, 19, 43, 33, 44, 45, 8, 9]. However, different from $l_\infty$ or $l_2$ constraint, it is a NP-hard problem to generate adversarial noise under $l_0$ constraint. To address this problem, many previous works have been proposed under both white-box and black-box setting. For white-box attack, JSMA [34] proposed to select the most effective pixels that influence the model decision based

---

[*]Work done during an internship at Microsoft Research Asia.
[†]Dongdong Chen is the corresponding author.

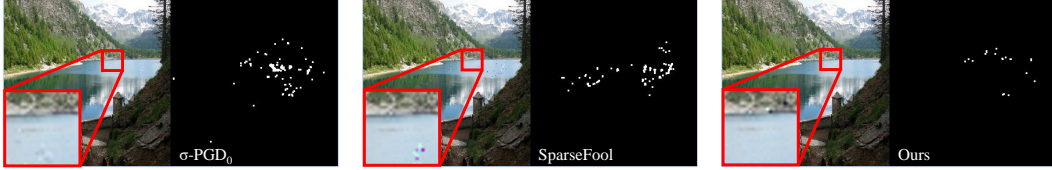

Figure 1: Adversarial samples and their corresponding perturbations generated by different sparse adversarial attack methods. Our method achieves the best performance of sparsity and invisibility.

on a saliency map. SparseFool [30] converted the problem into an $l_1$ constraint problem, $PGD_0$ [11] proposed to project the adversarial noise generated by PGD [27] to the $l_0$-ball to achieve the $l_0$ version PGD. When it comes to black-box attack, One Pixel Attack [39] and Pointwise Attack [37] proposed to apply evolutionary algorithm to achieve extremely sparse perturbations. LocSearchAdv [32] use the local search to realize sparse attack. Recently, CornerSearch [11] propose traversing all the pixels and selects the most effective subset of pixels to realize attack. In this paper, we focus on the white-box attack.

However, these existing methods still suffer from many issues. JSMA [34] is suffering from high complexity that can hardly achieve adversarial attacks on high resolution images. SparseFool [30] is not sparse enough and can not achieve a target attack. For $PGD_0$ [11], the number of perturbation pixels is defined before the attack and these pixels must be used for the attack, so it perturbs many redundant pixels and is probably not flexible for real scenarios.

To address these issues, we propose a novel greedy-based method dubbed as "GreedyFool". It mainly consists of two stages. In the first stage, we iteratively choose $k$ most suitable pixels to modify based on the gradient information until a successful attack. In the second stage, we also apply a greedy strategy to drop as many less important pixels as possible to further improve the sparsity.

On the other hand, we notice that sparse attacks are usually clearly visible as previous works presented [30, 11]. Therefore, we introduce a distortion map, which is an invisibility constraint that helps sparse attacks to achieve better invisibility. Instead of using hand-crafted distortion [11] which may ignore the semantic and content information in the image, we propose a framework based on Generative Adversarial Nets(GANs) [16] to learn a precise distortion map of the image. Such a distortion map will be added into the first stage of the "GreedyFool" as positional guidance.

Experiments on the CIFAR10 [22] and ImageNet [12] dataset show that the sparsity of our method is much better than state-of-the-art methods. For images in the ImageNet, when the perturbation threshold is set to 10, our method only needs to perturb 222.5 pixels while SparseFool needs to perturb 615 pixels, about $3\times$ more than our method. When relaxing the threshold to 255, our method only needs to modify 27 pixels to achieve an adversarial attack, while SparseFool needs 80.50 pixels. Meanwhile, our method demonstrates a higher target attack success rate than state-of-the-art methods. With a 200 pixels (0.22% pixels) budget on ImageNet, the target attack success rate of our GreedyFool is 15.52%, while $PGD_0$ is only 5.56%. When the budget increases to 1000 pixels (1.1% pixels), the success of our GreedyFool increases to 84.86%, while it is only 64.20% for $PGD_0$.

To summarize, the main contributions of this paper are threefold: 1) We propose a novel two-stage greedy-based adversarial attack method to further increase the sparsity. 2) To generate adversarial samples with better invisibility, we leverage GAN to generate a distortion cost map as the guidance to find proper pixels to modify. 3) Extensive experiments have demonstrated the superb performance of our method. For both non-target attack and target-attack, our method outperforms previous methods by large margins.

## 2 Distortion-Aware Sparse Adversarial Attack

**Problem Analysis.** We denote $\mathbf{x}$ as the source image and $y$ as its corresponding ground-truth label. Let $\mathcal{H}$ be the target model, $\mathcal{H}(\mathbf{x})_c$ is the output logit value for class $c$. For a clean input $\mathbf{x}$, $\operatorname{argmax}_c \mathcal{H}(\mathbf{x})_c = y$. An adversarial sample $\mathbf{x}^{adv} = \mathbf{x} + \mathbf{r}$ is generated by adding noise $\mathbf{r}$ to the original image $\mathbf{x}$ and satisfies $\operatorname{argmax}_c \mathcal{H}(\mathbf{x}^{adv})_c \neq y$. Meanwhile, the adversarial noise $\mathbf{r}$ should be small enough to guarantee the adversarial sample is similar to the original one. In most cases, it is

measured by a $l_p$-norm. In this paper, under the $l_0$ constraint, it should be

$$min_{\mathbf{r}}|\mathbf{r}|_0 \text{ subject to } argmax_c \mathcal{H}(\mathbf{x}^{adv})_c \neq y \tag{1}$$

However, it is a NP-hard problem to solve. So we resort to a greedy algorithm to find a local optimal result with fast speed. More specifically, we select $k$ current optimal pixels in each iteration based on the gradient information until a successful attack. But local optimal is not always the global optimal. To get better sparsity, we apply the greedy search again to find the unnecessary points in the selected points sets. Meanwhile, for better invisibility, we use a distortion value to instruct whether a pixel is suitable for modifying or not. Rather than a hand-crafted definition of each pixel's distortion, we propose using a generative adversarial network to learn a proper probability definition.

## 2.1 GreedyFool Adversarial Attack.

In the first stage, we iteratively increase the number of perturbed pixels until we find an adversarial sample. For clarity, we use a binary mask $\mathbf{m}$ to denote whether a pixel is selected and initialize $\mathbf{m}$ with all zeros. In each iteration, we run a forward-backward pass with the latest modified image $\mathbf{x}_t^{adv}$ to calculate the gradient of the loss function to $\mathbf{x}_t^{adv}$. And the pixels with bigger gradient values are regarded to contribute more adversary.

$$\mathbf{g}_t = \nabla_{\mathbf{x}_t^{adv}} \mathcal{L}(\mathbf{x}_t^{adv}, y, \mathcal{H})$$
$$\mathcal{L}(\mathbf{x}, y, \mathcal{H}) = max(\mathcal{H}(\mathbf{x})_y - max_{i \neq y}\{\mathcal{H}(\mathbf{x})_i\}, -\kappa) \tag{2}$$

where $\mathcal{L}(\mathbf{x}_t^{adv}, y, \mathcal{H})$ is the loss function and $\nabla_{\mathbf{x}_t^{adv}} \mathcal{L}(\mathbf{x}_t^{adv}, y, \mathcal{H})$ is the gradient of $\mathcal{L}(\mathbf{x}_t^{adv}, y, \mathcal{H})$ with respect of $\mathbf{x}_t^{adv}$. Here we use the loss function from C&W [6] for non-target attack. $\kappa$ is a confidence factor to control the attack strength, we set $\kappa = 0$ by default and enlarge it for better black-box transferability.

To achieve invisibility of the adversarial samples, we introduce a distortion map of an image, where the distortion of a pixel represents the visibility for the modification of the pixel, a higher distortion means the pixel modification can be more easily observed. We will introduce how to get the distortion map in the next section. Suppose the distortion map is $\varrho$, where $\varrho \in (0, 1)^{H \times W}$. To achieve a balance between invisibility and sparsity, we calculate a perturbation weight map $\mathbf{p}$ from $\varrho$ and select pixels with both $\mathbf{p}$ and $\mathbf{g}_t$. Formally,

$$p_{i,j} = \begin{cases} 0 & \varrho_{i,j} \geqslant \tau_1 \\ (\tau_1 - \varrho_{i,j})/(\tau_1 - \tau_2) & \tau_2 < \varrho_{i,j} \leqslant \tau_1 \\ 1 & \varrho_{i,j} \leqslant \tau_2 \end{cases} \tag{3}$$

$$\mathbf{g}_t' = \mathbf{p} \cdot \mathbf{g}_t \cdot (1 - \mathbf{m}) \tag{4}$$

where $\tau_1$ and $\tau_2$ are predefined thresholds and set as the 70, 25 percentile of $\varrho$ by default. Then we add the top $k$ unselected pixels which contain biggest $\mathbf{g}_t'$ values into the target perturbation pixels set and change corresponding values of $\mathbf{m}$ into 1. Finally, we update $\mathbf{x}_t^{adv}$ based on $\mathbf{g}_t$ and $\mathbf{m}$ with the perturb step size $\alpha$.

$$\mathbf{x_{t+1}^{adv}} = Clip_{\mathbf{x}}^{\epsilon}(\mathbf{x_t^{adv}} + \alpha \cdot \frac{\mathbf{g}_t \cdot \mathbf{m}}{|\mathbf{g}_t \cdot \mathbf{m}|}) \tag{5}$$

Besides the distortion map guidance, there are another two key implementation differences in our method when compared to existing sparse adversarial attack methods [30]. First, we search the sparse adversarial noise in a 'greedy' way for every iteration by using the latest updated gradient, rather than use an imitate direction estimated by other adversarial methods in SparseFool [30]. This ensures that every choice at the current iteration is the best and achieves an attack in earlier iterations. Second, SparseFool [30] think the adversary contribution of each pixel is linear additive, so it updates each selected pixel only once, while we view it as a non-linear process and update all the selected pixels in each iteration. This also helps us to achieve attack faster. Meanwhile, we notice that sparse attack is sensitive to the direction, so we keep both magnitude and sign of the gradient rather than only use its sign on each pixel. This makes the perturbation more precise and helps us achieve an attack with fewer pixels than previous methods. In the following experiments, we will show the adversary of perturbation is sensitive to its direction when its dimension is small.

**Algorithm 1** GreedyFool

**Input**: Source image $\mathbf{x}$, target model $\mathcal{H}$, distortion map $\varrho$.
**Parameter**: Max iterations $T$, threshold $\epsilon$, select number $k$.
**Output**: adversarial sample $\mathbf{x}^{adv}$

1: **Stage 1: Increasing**
2: Initialize $\mathbf{m} \leftarrow \mathbf{0}$, $t \leftarrow 0$, $\alpha = \epsilon/2$.
3: Initialize $\mathbf{x}_t^{adv} \leftarrow \mathbf{x}$
4: Generate perturbation weight map $\mathbf{p}$ from $\varrho$ with Equation. 3 .
5: **while** $t < T$ and $\mathbf{x}_t^{adv}$ is not adversarial **do**
6:     $\mathbf{g}_t \leftarrow \nabla_{\mathbf{x}_t^{adv}} L(\mathbf{x}_t^{adv}, y, \mathcal{H})$
7:     $\mathbf{g}_t' \leftarrow \mathbf{p} \cdot \mathbf{g}_t \cdot (1 - \mathbf{m})$
8:     $d_1, d_2, ..., d_k \leftarrow \mathrm{argtop_k}(|\mathbf{g}_t'|)$
9:     $m_{d_1, d_2, ..., d_k} = 1$
10:    $\mathbf{x}_{t+1}^{adv} = Clip_{\mathbf{x}}^{\epsilon}(\mathbf{x}_t^{adv} + \alpha \cdot \frac{\mathbf{g}_t \cdot \mathbf{m}}{|\mathbf{g}_t \cdot \mathbf{m}|})$
11:    $t \leftarrow t + 1$
12: **end while**
13:
14: **Stage 2: Reducing**
15: $R = \{\}$, $\mathbf{x}^{adv} \leftarrow \mathbf{x}_t^{adv}$, $\mathbf{r} \leftarrow \mathbf{x}^{adv} - \mathbf{x}$
16: **while** $t < T$ and $\{d | d \in \mathbb{N}, r_d \neq 0\} - R \neq \emptyset$ **do**
17:    $found = $ False.
18:    $d \leftarrow \underset{d \notin R, r_d \neq 0}{\mathrm{argmin}} |r_d|$
19:    $\mathbf{r}' \leftarrow \mathbf{r}$, $r_d' \leftarrow 0$.
20:    **for** $\alpha' = 1$ to $\epsilon$ and not $found$ **do**
21:       $\hat{\mathbf{x}}^{adv} = Clip_{\mathbf{x}}^{\epsilon}(\mathbf{x} + \alpha' \cdot \frac{\mathbf{r}'}{|\mathbf{r}'|})$
22:       **if** $\hat{\mathbf{x}}^{adv}$ is adversarial **then**
23:         $\mathbf{r} \leftarrow \mathbf{r}'$, $\mathbf{x}^{adv} \leftarrow \hat{\mathbf{x}}^{adv}$
24:         $found = $ True.
25:       **end if**
26:    **end for**
27:    **if** not $found$ **then**
28:       $R \leftarrow R \cup \{d\}$
29:    **end if**
30:    $t \leftarrow t + 1$
31: **end while**
32: **return** $\mathbf{x}^{adv}$

After the first stage, though the selected pixels can already ensure a successful sparse attack, we find some pixels can be redundant because of the greedy property. For a more sparse perturbation, it is necessary to minimize the number of perturbed pixels as small as possible. Therefore, we adopt a second reducing stage to achieve better sparsity.

Formally, we maintain a subset $R$ that contains the coordinate of the noise components which are necessary to the adversary and initialize it as an empty set. We suppose the pixels having smaller perturbation values may contribute less adversary than other pixels. So in each iteration, we choose the pixel with the minimum perturbation value from the subset $\{|r_d| | r_d \in \mathbf{r}, r_d \neq 0, d \notin R\}$ and drop it, then test the adversary of images generated with the dropped noise and a series of step size $\alpha'$ from 1 to the threshold $\epsilon$. If at least one of them is still an adversarial sample, we choose the adversarial sample which has the smallest $\alpha'$ as our new adversarial sample. If all the generated images are not adversarial samples, we regard the selected pixel as a necessary adversary component and add its coordinate into $R$. It should be noted that the forward pass of different threshold $\alpha'$ can be combined in a batch, so this process is very fast.

Meanwhile, our method can be viewed as a $l_0$ derivative of gradient-based methods, so for a given target label $tar$, we can achieve target attack by simply replacing the loss function in Eq. 2 with

$$\mathcal{L}_{tar}(\mathbf{x}, tar, \mathcal{H}) = max(max_{i \neq tar}\{\mathcal{H}(\mathbf{x})_i\} - \mathcal{H}(\mathbf{x})_{tar}, -\kappa) \tag{6}$$

The whole attack process is shown in Algorithm 1.

## 2.2 Distortion Map Generation via Generative Adversarial Networks

The purpose of distortion map is to evaluate the modification visibility to each pixel of an input image. A pixel with high distortion means the visibility is high when modifying it. Previous method [11] propose a hand-crafted $\sigma$-map to represent it. However, the $\sigma$-map only takes a $3 \times 3$ local patch into consideration and is highly frequency-related, which may ignore the content and semantic information of the image. To address this issue, we propose a new GAN-based framework for distortion map generation. The framework contains a generator $G$ and a discriminator $D$, as shown in Fig. 2. The generator $G$ and the discriminator $D$ plays a minimax game to get the distortion map.

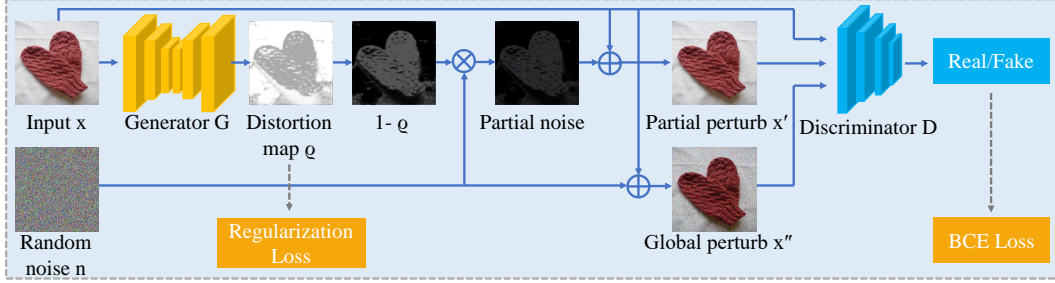

Figure 2: Illustration of our proposed GAN-based framework for distortion map generation. We calculate the partial perturb image $\mathbf{x}'$ and global perturb image $\mathbf{x}''$ for training. The BCE loss and an extra regularization loss are used during the training process.

For a given input $\mathbf{x}$ with size $H \times W \times C$, the distortion map is generated with the generator $G$: $\varrho = G(\mathbf{x})$ that $\varrho \in (0, 1)^{H \times W}$. With distortion map $\varrho$, we calculate the perturbed image $\mathbf{x}'$ as

$$\mathbf{x}' = \text{Clip}((1 - \varrho) \cdot \mathbf{n} + \mathbf{x}) \qquad (7)$$

where $\mathbf{n}$ is a random noise sampled from uniform distribution $(-\delta, \delta)$. $\delta$ is a predefined threshold and we set $\delta = 8/255$ in our experiments.

For the discriminator $D$, it tries to distinguish the real image $\mathbf{x}$ and perturbed image $\mathbf{x}'$. In order to make the GAN training more stable, we also build a globally perturbed image $\mathbf{x}'' = \mathbf{x} + \mathbf{n}$ and add it for training discriminator D. So the loss for discriminator D is:

$$\mathcal{L}_D(\mathbf{x}) = -(2\log(D(\mathbf{x})) + \log(1 - D(\mathbf{x}')) + \log(1 - D(\mathbf{x}''))) \qquad (8)$$

The target of the generator $G$ is to generate a proper distortion map $\mathbf{p}$ that the perturbed image $\mathbf{x}'$ and original $\mathbf{x}$ is indistinguishable for $D$, so the generator $G$ will encourage $\varrho = \mathbf{1}^{H \times W}$. In this situation, no modification will be made to $\mathbf{x}$. So we add a regularization loss to the distortion map which requires the distortion map to be as smaller as possible. This regularization loss will force the generator to learn to add perturbations to the input image and make this perturbed image indistinguishable by the discriminator. The location on the image where perturbations can be added should have a low visibility for discrimination, which indicates the small value of $\varrho$. The overall loss for generator is:

$$\mathcal{L}_G(\mathbf{x}) = -\log(D(\mathbf{x}')) + \lambda|\varrho|, \qquad (9)$$

where $\lambda$ is loss weight for the regularization loss, we choose $1e - 5$ in our experiment by default. Model architecture and training detail please refer to the supplementary materials.

## 3 Experiments

### 3.1 Sparsity Evaluation

We first evaluate the sparsity under different perturbation thresholds. On the CIFAR10 [22] dataset, we compare our methods with recent works: JSMA [34], SparseFool [30], and $\text{PGD}_0$ [11] ($\sigma$-$\text{PGD}_0$ for settings $\epsilon \neq 255$). For the ImageNet [12] dataset, since JSMA is too slow to achieve adversarial attack for large images, we only compare with SparseFool and $\text{PGD}_0$. In the following experiments, we use the official implementation of SparseFool [1] and $\text{PGD}_0$ [2] and follow their default settings. For JSMA, we use the implementation from FoolBox2.4 [3]. For our GreedyFool, we set the select number $k = 1$ when $\epsilon \geqslant 128$. When $\epsilon < 128$, we initialize $k$ with 1 and increase 1 after each iteration for faster speed. As we introduced in Sec. 1, $\text{PGD}_0$ needs a pre-defined number of perturbation pixels and can only calculate the fooling rate under such a pre-defined number(we name it as static evaluation and report it as the $m$ pixels fooling rate in the following). While other methods perturb image with a dynamic pixel number and run until a successful attack or iteration upper bound, the evaluation metric of these methods is the mean and median of the modified pixel numbers( we name it as dynamic evaluation and report it with mean, median of perturbed pixel number, and total fooling rate in the following). For a fair comparison, we report the result on both metrics. In the following

Table 1: Non-target attack sparsity comparison on ImageNet dataset. $m$ Pixels Fooling Rate means the fooling rate when only allow to perturb at most $m$ pixels.

| Threshold | Method | Dynamic Evaluation | | | Static Evaluation | | | | |
|---|---|---|---|---|---|---|---|---|---|
| | | Mean | Median | Fooling Rate(%) | \multicolumn{5}{c}{$m$ Pixels Fooling Rate(%)} | | | | |
| | | | | | 10 | 20 | 50 | 100 | 200 |
| $\epsilon = 255$ | $PGD_0$ | - | - | - | 28.26 | 38.77 | 61.11 | 78.79 | 92.61 |
| | $\sigma$-$PGD_0$ | - | - | - | 17.52 | 22.56 | 37.78 | 48.13 | 61.88 |
| | SparseFool | 147.61 | 80.50 | 100.00 | 16.10 | 25.03 | 38.08 | 56.04 | 75.22 |
| | **GF(ours)** | **62.13** | **27.00** | 100.00 | **29.60** | **42.65** | **67.24** | **82.87** | **94.60** |
| | | | | | 10 | 20 | 50 | 100 | 200 |
| $\epsilon = 100$ | $\sigma$-$PGD_0$ | - | - | - | 12.76 | 17.97 | 25.78 | 36.29 | 45.12 |
| | SparseFool | 172.24 | 94.00 | 100.00 | 14.22 | 20.60 | 35.82 | 52.00 | 77.21 |
| | **GF(ours)** | **89.27** | **41.00** | 100.00 | **22.76** | **34.36** | **56.61** | **73.70** | **88.28** |
| | | | | | 50 | 100 | 200 | 500 | 1000 |
| $\epsilon = 10$ | $\sigma$-$PGD_0$ | - | - | - | 5.71 | 9.29 | 13.75 | 21.19 | 28.43 |
| | SparseFool | 1061.74 | 615.00 | 100.00 | 10.80 | 17.82 | 25.02 | 43.20 | 65.25 |
| | **GF(ours)** | **552.11** | **222.50** | 100.00 | **21.13** | **32.46** | **47.20** | **70.59** | **85.57** |

Table 2: Non-target attack sparsity comparison on CIFAR10 dataset. $m$ Pixels Fooling Rate means the fooling rate when only allow to perturb at most $m$ pixels.

| Threshold | Method | Dynamic Evaluation | | | Static Evaluation | | | | |
|---|---|---|---|---|---|---|---|---|---|
| | | Mean | Median | Fooling Rate(%) | \multicolumn{5}{c}{$m$ Pixels Fooling Rate(%)} | | | | |
| | | | | | 1 | 2 | 5 | 10 | 20 |
| $\epsilon = 255$ | JSMA | 18.51 | 13 | 100.00 | 4.81 | 9.67 | 23.59 | 42.06 | 67.84 |
| | $PGD_0$ | - | - | - | **15.22** | 25.82 | 50.93 | 75.46 | 93.73 |
| | $\sigma$-$PGD_0$ | - | - | - | 7.32 | 10.00 | 22.24 | 40.93 | 64.78 |
| | SparseFool | 17.29 | 13 | 100.00 | 6.52 | 11.31 | 23.88 | 43.10 | 69.32 |
| | **GF(ours)** | **5.98** | **4** | 100.00 | 12.41 | **29.06** | **56.26** | **79.30** | **95.77** |
| | | | | | 5 | 10 | 15 | 20 | 30 |
| $\epsilon = 100$ | $\sigma$-$PGD_0$ | - | - | - | 11.71 | 22.17 | 29.17 | 33.84 | 43.28 |
| | SparseFool | 18.99 | 14 | 100.00 | 20.30 | 38.25 | 52.81 | 65.74 | 80.47 |
| | **GF(ours)** | **10.53** | **8** | 100.00 | **42.01** | **67.13** | **81.86** | **89.99** | **97.13** |
| | | | | | 30 | 40 | 50 | 60 | 100 |
| $\epsilon = 10$ | $\sigma$-$PGD_0$ | - | - | - | 5.58 | 7.20 | 7.87 | 8.49 | 10.84 |
| | SparseFool | 163.54 | 128 | 100.00 | 15.78 | 19.94 | 25.53 | 27.04 | 41.07 |
| | **GF(ours)** | **47.82** | **34** | 100.00 | **42.26** | **52.17** | **60.27** | **67.70** | **85.39** |

experiments, we generate adversarial samples by attacking an Inception-v3 [40] model pretrained on ImageNet [12] dataset. For CIFAR10 dataset, we use a pretrained network in network(NIN) [25] model as [11]. To make an accurate comparison, we generate adversarial samples with 5000 images randomly selected from the ImageNet validation set and 10000 images from the CIFAR10 test set.

**Non-target Attack Result.** Non-target attack results on both ImageNet [12] and CIFAR10 dataset are shown in Table 1 and Table 2. For ImageNet, when $\epsilon = 255$, for dynamic evaluation, the median perturbation number of our GreedyFool is only 27(0.03%pixels) to achieve attack, while SparseFool needs to perturb 80.5 pixels, nearly $3\times$ more than us. As the threshold $\epsilon$ decreases, the perturbation number needed for both methods increases, while our method is still better than SparseFool with large margins. When it comes to static evaluation, our GreedyFool still has the best performance. We also notice the fooling rate of $\sigma$-$PGD_0$ is much lower than $PGD_0$, when $m = 100$ and $\epsilon = 255$, the fooling rate of $PGD_0$ is 78.79%, while $\sigma$-$PGD_0$ is only 48.13%. In the meantime, the fooling rate of our GreedyFool is 82.87%.

When it comes to the CIFAR10 dataset, we find that our method still outperforms other methods in most settings. The only exception is the static evaluation with $m = 1$ and $\epsilon = 255$ and this is due to the restart strategy of $PGD_0$. But as $m$ increases, the fooling rate of $PGD_0$ increases slowly, while our GreedyFool increases rapidly and becomes better than $PGD_0$.

**Target Attack Result.** As SparseFool can only achieve non-target attack, here we compare target attack results with $PGD_0$ on both CIFAR and ImageNet dataset in Table. 4. Though target attack

Table 3: Speed comparison on both the CIFAR10 and ImageNet dataset.

| Method | JSMA | $PGD_0$ | $\sigma$-$PGD_0$ | SparseFool | GreedyFool(ours) |
|---|---|---|---|---|---|
| CIFAR10 | 1.192 | 0.169 | 0.179 | 0.291 | **0.114** |
| ImageNet | - | **3.232** | 3.731 | 8.380 | 5.253 |

Table 4: Target attack success rate on both CIFAR10 and ImageNet dataset. $m$ Pixels Fooling Rate means the fooling rate when only allow to perturb at most $m$ pixels

| Threshold | Method | CIFAR10 $m$ pixels Fooling Rate(%) | | | | | ImageNet $m$ Pixels Fooling Rate(%) | | | | |
|---|---|---|---|---|---|---|---|---|---|---|---|
| | | 1 | 10 | 20 | 50 | 100 | 200 | 400 | 1000 | 2000 | 4000 |
| $\epsilon = 255$ | $PGD_0$ | 1.38 | 30.94 | 63.88 | 96.50 | 99.44 | 5.56 | 22.50 | 64.20 | 83.89 | 93.10 |
| | $\sigma$-$PGD_0$ | 0.67 | 9.10 | 22.13 | 55.18 | 74.33 | 0.65 | 1.20 | 5.45 | 15.55 | 25.52 |
| | **GF(ours)** | **2.21** | **32.91** | **67.49** | **98.21** | **99.98** | **15.52** | **43.63** | **84.86** | **93.82** | **98.00** |
| $\epsilon = 100$ | $\sigma$-$PGD_0$ | 0.33 | 3.05 | 5.59 | 19.17 | 35.77 | 0.00 | 0.18 | 0.37 | 2.22 | 6.82 |
| | **GF(ours)** | **1.53** | **21.36** | **48.78** | **93.09** | **99.88** | **9.21** | **30.80** | **84.29** | **95.88** | **99.20** |
| $\epsilon = 10$ | $\sigma$-$PGD_0$ | 0.00 | 0.16 | 0.39 | 0.50 | 0.94 | 0.10 | 0.25 | 0.15 | 0.15 | 0.25 |
| | **GF(ours)** | **0.14** | **2.49** | **5.62** | **19.27** | **45.23** | **0.20** | **1.60** | **8.20** | **31.24** | **70.63** |

is much harder than non-target attack, our GreedyFool outperforms $PGD_0$ and $\sigma$-$PGD_0$ by a large margin. Besides, we find it is hard for $\sigma$-$PGD_0$ to find target-attack adversarial samples. For example, even when the pixel budget $m = 4000$ and $\epsilon = 255$, its fooling rate is only 25.52%.

**Speed comparison.** Here we compare the speed of our GreedyFool with state-of-the-art methods in Table 3. When the input image is small, all the methods can achieve attack quickly and our GreedyFool only needs 0.114 seconds on average. On the ImageNet dataset, our GreedyFool can still achieve comparable speed to other methods while guaranteeing better sparsity.

**Black-box Transfer Attack Result.** As we introduced in Sec2.1, $\kappa$ controls the attack strength that when $\kappa = 0$, we stop our attack once the generated adversarial sample is adversary. When $\kappa > 0$, we keep adding pixels to increase the attack confidence until the logit difference $max_{i \neq y}\{\mathcal{H}(\mathbf{x})_i\} - \mathcal{H}(\mathbf{x})_y > \kappa$. It is obvious that $\kappa$ is contradiction with the Reduce stage of our GreedyFool, so we only use the Increasing stage of GreedyFool to evaluate the influence of $\kappa$. Here we use the black-box transferability to evaluate the adversary that higher fooling rate indicates better adversary. In this section, we compare with SparseFool on the ImageNet dataset. Adversarial samples are generated by attacking a DenseNet161 [21] model with $\epsilon = 255$ and test the fooling rate on VGG16 [38] and Resnet50 [20] model, all the models are pretrained on ImageNet dataset.

Results are reported in Table.5, we find that when $\kappa = 0$, the black-box fooling rates of SparseFool are 15.67% and 26.00% respectively, similar with the result of our GreedyFool which is 17.00% and 23.33%, while the median pixel perturbation number of our GreedyFool is only 32.00, nearly 3.5× smaller than SparseFool. With the increase of $\kappa$, both the perturbed pixels number and the transferability increases. When $\kappa = 6$, the transferability of our methods is nearly 2× better than SparseFool, while the perturbed pixel number is still smaller than it. The above results prove that our GreedyFool is efficient and flexible. More results please refer to the supplemental material.

## 3.2 Invisibility Evaluation.

For adversarial samples, invisibility is an important property. It can be analyzed from two aspects: invisibility to human eyes and invisibility to machines. Since the adversarial perturbations generated by existing methods are already very small, human invisibility is easy to achieve. But for machine invisibility, it is much more challenging. There are many recent works [5, 7, 46, 28, 29] that use powerful classification models to detect adversarial samples. In this paper, we consider two statistics-based methods as the metric of machine invisibility. The first is the state-of-the-art steganalysis based adversarial detection method SRM [26]. Another is trying to train a powerful binary CNN classifier to separate the generated adversarial samples from clean images. The underlying motivation is that, if the adversarial samples are invisible enough, the classifier will be difficult to converge and results in random guess (50% accuracy). More details are given in the supplementary materials.

In Table 6, we compare our method not only with sparse attack methods but also one strong $l_2$ baseline C&W [6] and $l_\infty$ baseline I-FGSM [23]. For C&W, we use 4 search steps and 25/100

Table 5: Black-box transferability on ImageNet dataset. $\kappa$ is the confidence factor used in our loss function Eq.2. Here FR denote the fooling rate and * means whie-box attack results.

| Method | Mean | Median | FR DenseNet161(%) | FR ResNet50(%) | FR VGG16(%) |
|---|---|---|---|---|---|
| SparseFool | 187.39 | 111.00 | 100.00* | 15.67 | 26.00 |
| GreedyFool ($\kappa = 0$) | **47.79** | **32.00** | 100.00* | 17.00 | 23.33 |
| GreedyFool ($\kappa = 1$) | 59.06 | 41.00 | 100.00* | 18.00 | 29.00 |
| GreedyFool ($\kappa = 2$) | 72.37 | 53.50 | 100.00* | 23.33 | 29.67 |
| GreedyFool ($\kappa = 3$) | 85.37 | 65.50 | 100.00* | 24.67 | 35.67 |
| GreedyFool ($\kappa = 4$) | 99.57 | 75.00 | 100.00* | 27.67 | 40.00 |
| GreedyFool ($\kappa = 5$) | 113.43 | 85.50 | 100.00* | 30.33 | 45.00 |
| GreedyFool ($\kappa = 6$) | 127.48 | 97.00 | 100.00* | **32.33** | **48.00** |

Table 6: Machine invisibility comparison with the SRM detection rate and binary CNN classifier Accuracy metric. All adversarial samples are generated by non-target attacking an pretrained Inception-v3 model with image size $299 \times 299$.

| Attack Method | Perturbation | | | Fooling Rate(%) | SRM DetRate(%) | Classifier Acc(%) |
|---|---|---|---|---|---|---|
| | $L_0$ | $L_2$ | $L_\infty$ | | | |
| I-FGSM($\epsilon = 1$) | 80372.00 | 0.78 | 1.00 | 94.60 | 99.42 | 86.15 |
| I-FGSM($\epsilon = 2$) | 83420.00 | 1.29 | 2.00 | 98.60 | 99.64 | 96.43 |
| I-FGSM($\epsilon = 4$) | 87870.05 | 2.46 | 4.00 | 100.00 | 99.66 | 98.74 |
| C&W $4 \times 25$ | 78018.50 | 0.97 | 2.00 | 94.81 | 99.52 | 87.59 |
| C&W $4 \times 100$ | 26943.00 | 0.46 | 4.99 | 100.00 | 94.80 | 78.84 |
| PGD$_0$ ($\epsilon = 255$) | 200.00 | 7.13 | 255.00 | 92.61 | 87.80 | 99.90 |
| $\sigma$-PGD$_0$ ($\epsilon = 255$) | 1000.00 | 3.13 | 113.92 | 90.98 | 79.00 | 94.90 |
| SparseFool ($\epsilon = 255$) | 81.00 | 2.87 | 255.00 | 100.00 | 74.36 | 97.66 |
| **GF(ours)**($\epsilon = 255$) | 27.00 | 2.66 | 255.00 | 100.00 | 61.54 | 94.02 |
| SparseFool ($\epsilon = 10$) | 642.00 | 0.59 | 10.00 | 100.00 | 73.58 | 86.79 |
| **GF(ours)**($\epsilon = 10$) | 222.50 | 0.51 | 10.00 | 100.00 | 54.00 | 67.90 |
| **GF(ours)**($\epsilon = 5$) | 472.00 | 0.36 | 5.00 | 100.00 | 54.32 | 59.78 |
| **GF(ours)**($\epsilon = 2.5$) | 1121.00 | 0.28 | 2.50 | 100.00 | 57.24 | 50.00 |

iterations for each step, and we denote it as C&W $4 \times 25$ and C&W $4 \times 100$ respectively. When $\epsilon = 255$, we set $l_0 = 200$ for PGD$_0$ and $l_0 = 1000$ for $\sigma$-PGD$_0$ to ensure the success rate higher than 90%. When $\epsilon = 10$, even we set $l_0 = 80000$, the success rate is still only 73.56% so we do not report it here. We find that I-FGSM with $\epsilon = 4$ is the easiest to be detected as it uses a relatively large perturbation threshold and nearly perturbs all the pixels. Consistent with the observation in [26], SRM is very sensitive to small perturbation. Even for I-FGSM($\epsilon = 1$), its detection rate is still larger than 99%. For C&W, since the perturbations on some pixels are smaller than 1 and removed by the round operation when the image is saved, it is relatively sparse and has a slightly lower detection rate.When it comes to $l_0$-based methods, as the SRM is specially designed for detecting small while dense perturbations, it is not sensitive to large but sparse noise. For example, for SparseFool with $\epsilon = 255$, even if it is obvious to a human, its detection rate is only 74.36%. For PGD$_0$ and $\sigma$-PGD$_0$, suffered by its redundant perturbed pixels, its detection rate is 87.80% and 79.00% respectively. For our GreedyFool, with the distortion map guidance, the detection rate is much lower with 61.54%.

For the classifier-based metric, it works well for most $l_2, l_\infty$ methods. For sparse attack methods with $l_0$ constraint, when the perturbation is large ($\epsilon = 255$), training a CNN binary classifier can achieve 97.66% accuracy for SparseFool and 94.02% for GreedyFool. While PGD$_0$ and $\sigma$-PGD$_0$ also performs poorly for the same reason. As $\epsilon$ decreases, the adversarial classification accuracy decreases but our GreedyFool performs better. For example, when $\epsilon = 10$, the classification accuracy of SparseFool is 86.79%, but our GreedyFool is only 67.90%. Especially when $\epsilon = 2.5$, the accuracy decreases to 50.00% with random guess.

### 3.3 Ablation Study

**Component Analysis of GreedyFool.** We further analyze the contributions of each part of our method toward sparsity and invisibility. In the following, we denote the increasing stage as 'Incr', then evaluate the effect of the reducing stage ('Reduce') and the introduced distortion map ('Dis'). Here we set $\epsilon = 10$ and use the binary CNN classifier for invisibility evaluation.

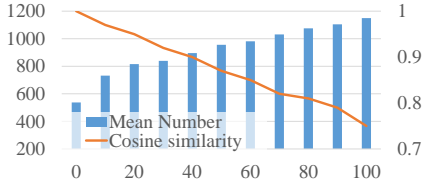

Figure 3: The influence of gradient direction by changing different percentiles of gradient into sign.

Table 7: The contribution of each part in GreedyFool, here FR and Acc denote the fooling rate and classifier accuracy.

| Method | Mean | Median | FR(%) | Acc(%) |
|---|---|---|---|---|
| Incr | 537.10 | 188.00 | 100 | 75.65 |
| Incr + Reduce | 426.77 | 161.00 | 100 | 75.35 |
| Incr + Dis | 607.22 | 243.50 | 100 | 68.50 |
| Incr + Reduce +Dis | 552.11 | 222.50 | 100 | 67.90 |

From the results shown in Table 7, we observe that with only the 'Incr' part, our method already reaches state-of-the-art sparsity with a modification number of 188. Meanwhile, the classifier accuracy is relatively low(similar with C&W $4 \times 100$ in Table 6). We believe such a low accuracy benefits from the extremely low perturbation number. When we add the 'Reduce' stage, the mean modification number decreases from $537.10$ to $426.77$. This proves our hypothesis that the result of the greedy strategy in the 'Incr' stage is not optimal. As the modification number decreases, we find the classifier accuracy also decreases a bit. When we add the 'Dis' part into our method as the modification position guidance, the modification number increases a lot, it is reasonable because some pixels which influence the adversary a lot but have a high distortion are abandoned. However, the classifier accuracy decreases from 75.65% to 68.50% significantly. This proves our idea that proper modification position is crucial to invisibility. Finally, By combining the 'Reduce' and 'Dis' part, our method gets a satisfying result for both sparsity and invisibility.

**Sign or Direction.** As we mentioned in Sec. 2.1, we find when the dimension of adversarial perturbation is small, its adversary is sensitive to its direction. In this section, we evaluate the influence of the direction change quantitatively.

Formally, we sort the non-zero noise from large to small, then we use the $q$-th percentile value in it to scale the noise and clamp it to $[0, 1]$, with such operation, we set $q$-th percents noise to its sign and scale other small noise components proportionally. Therefore, the direction of the new noise is the same as the original gradient direction when $q = 0$ and degrades to the sign direction when $q = 100$. Here we set $\epsilon = 10$ and evaluate the change of the direction by the cosine similarity in the following.

Figure 3 shows the influence of gradient direction change. We find that when $q = 0$, the mean modification number is only $537.10$. With the increase of $q$, the modification direction deviates from the original direction and the modification number increases. When $q = 100$, we find the cosine similarity to be only $0.75$ and the modification number increases to $1149.39$, nearly 2 times larger than the best case when $q = 0$.

This phenomenon is different from the results of previous dense attack methods, such as I-FGSM, that the performance difference between using sign or not is negligible. We propose a reasonable explanation from the infinitesimal accumulate idea proposed in [17]. [17] believes the existence of adversarial samples is because DNNs are not non-linear enough in the high dimension, so infinitesimal caused by the small perturbations accumulates linearly and finally changes the prediction. When it comes to global $l_\infty$ attack methods, even the sign operation changes the direction, a global perturbation (268203 dims for images with size $299 \times 299$ and 3 channels) still accumulates a large number of infinitesimal to change the prediction. But when it comes to our sparse attack, the dimension is much smaller (1611.3 dims for 537.1 perturbed pixels with 3 channels), so infinitesimal accumulated by each dimension is more important and the direction change is more sensitive.

## 4 Conclusion

In this paper, we propose a novel two-stage distortion-aware greedy-based sparse adversarial attack method "GreedyFool". It can achieve both much better sparsity and invisibility than existing state-of-the-art sparse adversarial attack methods. It basically first selects the most effective candidate positions to modify with gradient, then utilizes a reduce stage to drop some less important points. To get better invisibility, we propose using a GAN-based distortion map as guidance in the first stage. In the future, we will further investigate how to incorporate the proposed idea into $l_2$ and $l_\infty$ based adversarial attacks to help achieve better invisibility.

## Acknowledgement.

This work was supported in part by the Natural Science Foundation of China under Grant U1636201, 62002334 and 62072421, Exploration Fund Project of University of Science and Technology of China under Grant YD3480002001, and by Fundamental Research Funds for the Central Universities under Grant WK2100000011.

## Broader Impact.

Technically, the newly proposed method boosts existing sparse adversarial attacks with much better sparsity and invisibility, which is indeed big progress. For a broad deep learning community, it will help them understand the fragility of CNN more thoroughly and design more robust network structures.

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
