[Supplementary Material]

# GreedyFool: Distortion-Aware Sparse Adversarial Attack supplementary material

**Xiaoyi Dong**[1]*, **Dongdong Chen**[2]†, **Jianmin Bao**[2], **Chuan Qin**[1],
**Lu Yuan**[2], **Weiming Zhang**[1], **Nenghai Yu**[1], **Dong Chen**[2]
[1]University of Science and Technology of China [2]Microsoft Research
{dlight@mail., qc94@mail., zhangwm@, ynh@ }.ustc.edu.cn
cddlyf@gmail.com , {jianbao, luyuan, doch }@microsoft.com

## 1 Detail about Distortion Map Generation via GANs

For generator $G$, we use a simple architecture that contains two down sampling blocks, two residual blocks, two up sampling blocks, and ends with a $tanh$ function. For discriminator $D$ on ImageNet, we use an Inception V3 model for binary classification. When it comes to CIFAR10, we use a cifar-resnet18 model. We train the models with 45 epochs for both CIFAR10 and ImageNet dataset. For Generator $G$, we use the Adam optimizer with learning rate 1e-3. For Discriminator $D$, we use the Adamax optimizer with learning rate 1e-4. Meanwhile, we find that it is unnecessary to train GAN for each dataset. A GAN trained on ImageNet (GAN-ImageNet) also works on other datasets. If we generate the distortion map with GAN-ImageNet for CIFAR10 images by resizing, the mean perturbed pixel number is 6.01 and the median is still 4, almost the same as the result in our paper.

Fig.1 shows the result of our Distortion Map on the ImageNet dataset. Meanwhile, we also show the $\sigma$-map used in PGD$_0$ [2]. We find that the distortion map generated by our methods pays attention to the texture and foreground part of the image, and ignores the relatively smooth background part of the image. However, the $\sigma$-map is calculated within a 3×3 neighbor, so it calculates a non-zero value for the smooth but not absolute smooth part. This is different from our visual intuition that we should not perturb the relatively smooth part of the image.

Besides the above qualitative analysis, we further report the quantitative result by replacing the GAN-based distortion map with the $\sigma$-map and generate adversarial samples with $\epsilon = 10$. The median perturbation number and SRM detection rate of the $\sigma$-GreedyFool is 301.50 and 57.20%, which is worse than our GreedyFool which is 222.50 and 54.00% respectively.

## 2 More Attack Result

### 2.1 Sparse-only result

In the main experiments, we concern not only the sparsity, but also the invisibility. So the results of both our GreedyFool and other methods are the invisibility-concerned setting( distortion map for our GreedyFool and $\sigma$-map for the PGD$_0$). Here we report the results about only the sparsity. For GreedyFool, we remove the distortion map. For PGD$_0$, we use the PGD$_0$ $l_0 + l_\infty$. For SparseFool, we also use the $l_0 + l_\infty$ setting for it. Results on ImageNet are shown in Table.1 with the same setting with Table.1 in the main paper.

Table 1: Non-target attack sparsity comparison on ImageNet dataset. $m$ Pixels Fooling Rate means the fooling rate when only allow to perturb at most $m$ pixels.

| Threshold | Method | Dynamic Evaluation | | | Static Evaluation | | | | |
|---|---|---|---|---|---|---|---|---|---|
| | | Mean | Median | Fooling Rate(%) | $m$ Pixels Fooling Rate(%) | | | | |
| | | | | | 10 | 20 | 50 | 100 | 200 |
| | $PGD_0$ | - | - | - | 28.26 | 38.77 | 61.11 | 78.79 | 92.61 |
| | $\sigma$-$PGD_0$ | - | - | - | 17.52 | 22.56 | 37.78 | 48.13 | 61.88 |
| $\epsilon = 255$ | SparseFool | 147.61 | 80.50 | 100.00 | 16.10 | 25.03 | 38.08 | 56.04 | 75.22 |
| | **GreedyFool** | 62.13 | 27.00 | 100.00 | 29.60 | 42.65 | 67.24 | 82.87 | 94.60 |
| | **GreedyFool** w/o Dis | **53.07** | **25.00** | 100.00 | **31.76** | **45.72** | **67.77** | **84.80** | **94.97** |
| | | | | | 10 | 20 | 50 | 100 | 200 |
| | $PGD_0$ $l_0 + l_\infty$ | - | - | - | 19.60 | 27.45 | 39.70 | 55.40 | 76.34 |
| | $\sigma$-$PGD_0$ | - | - | - | 12.76 | 17.97 | 25.78 | 36.29 | 45.12 |
| $\epsilon = 100$ | SparseFool | 172.24 | 94.00 | 100.00 | 14.22 | 20.60 | 35.82 | 52.00 | 77.21 |
| | **GreedyFool** | 89.27 | 41.00 | 100.00 | 22.76 | 34.36 | 56.61 | 73.70 | 88.28 |
| | **GreedyFool** w/o Dis | **82.02** | **36.00** | 100.00 | **25.83** | **37.03** | **58.18** | **76.40** | **90.20** |
| | | | | | 50 | 100 | 200 | 500 | 1000 |
| | $PGD_0$ $l_0 + l_\infty$ | - | - | - | 22.77 | 30.82 | 41.05 | 56.76 | 68.69 |
| | $\sigma$-$PGD_0$ | - | - | - | 5.71 | 9.29 | 13.75 | 21.19 | 28.43 |
| $\epsilon = 10$ | SparseFool | 1061.74 | 615.00 | 100.00 | 10.80 | 17.82 | 25.02 | 43.20 | 65.25 |
| | **GreedyFool** | 552.11 | 222.50 | 100.00 | 21.13 | 32.46 | 47.20 | 70.59 | 85.57 |
| | **GreedyFool** w/o Dis | **426.77** | **161.00** | 100.00 | **26.75** | **38.83** | **55.76** | **79.23** | **91.01** |

We find that without the distortion map, the sparsity of our GreedyFool increase a bit. When it comes to $PGD_0$ $l_0 + l_\infty$, without the strong constrain of $\sigma$-map, it performs better than SparseFool, but still worse than our GreedyFool.

## 2.2 More Black-box results

In the main paper, we report part of the attack results on DenseNet161 and its corresponding black-box results. Here we show the results in Table.2. We find for all the three models, our GreedyFool outperforms SparseFool for both sparsity and transferability. The smaller perturbation number indicates that the pixel GreedyFool perturbed are more accurate. Meanwhile, we get the same conclusion with [4] that it is much easier to transfer from large model to small model.

## 3 Detail about Invisibility Evaluation

We selected 12000 images from the ImageNet validation set for generating adversarial samples. As the adversarial noise distribution and pattern are different between different adversarial attack methods, we train the classifier for each method respectively to achieve a more accurate detection rate. For each adversarial attack method, we use 10000 images to train the classifier and test the detection rate on the remaining 2000 images. Here we use the default setting in [3] for the SRM-based detector. For the CNN-based binary classifier, we use SRNet [1], which is designed for detecting small noise specifically and performances better than classics classification models such as Inception-v3 or ResNet. However, it is still hard to converge when the noise is small(such as C&W $4 \times 100$ and our GreedyFool with $\epsilon \leqslant 10$), so we finetune the model with a pretrained model. Specifically, for dense attack methods(I-FGSM and C&W), we first train the model on adversarial samples generated by I-FGSM($\epsilon = 4$) and finetune the model on adversarial samples generated by rest methods with it. For sparse attack methods, we train the model from scratch when the adversarial samples are generated with $\epsilon = 255$, and finetune the model with the model pretrained on SparseFool($\epsilon = 255$) when the adversarial samples are generated with $\epsilon \leqslant 10$. In details, the initial learning rates for pre-trained and fine-tuning are $1e - 3$ and $1e - 4$ respectively. The total training epoch is 300 and the learning rate is decayed by 0.1 when the epoch is 100, 200, and 275.

Table 2: Black-box transferability on ImageNet dataset. Here * means whie-box attack results.

| Model | Method | Mean | Median | DenseNet161(%) | ResNet50(%) | VGG16(%) |
|---|---|---|---|---|---|---|
| DenseNet161 | SparseFool | 187.39 | 111.00 | 100.00* | 15.67 | 26.00 |
| | GreedyFool ($\kappa = 0$) | **47.79** | **32.00** | 100.00* | 17.00 | 23.33 |
| | GreedyFool ($\kappa = 1$) | 59.06 | 41.00 | 100.00* | 18.00 | 29.00 |
| | GreedyFool ($\kappa = 2$) | 72.37 | 53.50 | 100.00* | 23.33 | 29.67 |
| | GreedyFool ($\kappa = 3$) | 85.37 | 65.50 | 100.00* | 24.67 | 35.67 |
| | GreedyFool ($\kappa = 4$) | 99.57 | 75.00 | 100.00* | 27.67 | 40.00 |
| | GreedyFool ($\kappa = 5$) | 113.43 | 85.50 | 100.00* | 30.33 | 45.00 |
| | GreedyFool ($\kappa = 6$) | 127.48 | 97.00 | 100.00* | **32.33** | **48.00** |
| ResNet50 | SparseFool | 151.52 | 77.50 | 9.67 | 100.00* | 21.33 |
| | GreedyFool ($\kappa = 0$) | **36.41** | **22.00** | 9.67 | 100.00* | 21.00 |
| | GreedyFool ($\kappa = 1$) | 47.40 | 32.50 | 13.00 | 100.00* | 26.33 |
| | GreedyFool ($\kappa = 2$) | 58.48 | 41.00 | 16.00 | 100.00* | 32.00 |
| | GreedyFool ($\kappa = 3$) | 71.41 | 53.50 | 17.33 | 100.00* | 36.00 |
| | GreedyFool ($\kappa = 4$) | 84.20 | 62.50 | 21.67 | 100.00* | 40.67 |
| | GreedyFool ($\kappa = 5$) | 94.27 | 71.00 | 21.33 | 100.00* | 44.00 |
| | GreedyFool ($\kappa = 6$) | 109.30 | 85.00 | **23.67** | 100.00* | **44.67** |
| VGG16 | SparseFool | 139.92 | 66.50 | 7.67 | 10.00 | 100.00* |
| | GreedyFool ($\kappa = 0$) | **24.42** | **15.50** | 7.00 | 9.00 | 100.00* |
| | GreedyFool ($\kappa = 1$) | 33.61 | 22.00 | 12.67 | 12.67 | 100.00* |
| | GreedyFool ($\kappa = 2$) | 41.76 | 30.50 | 14.00 | 16.33 | 100.00* |
| | GreedyFool ($\kappa = 3$) | 49.85 | 38.00 | 15.67 | 17.00 | 100.00* |
| | GreedyFool ($\kappa = 4$) | 60.17 | 46.50 | 16.00 | 17.67 | 100.00* |
| | GreedyFool ($\kappa = 5$) | 68.77 | 54.50 | 17.33 | 19.33 | 100.00* |
| | GreedyFool ($\kappa = 6$) | 77.62 | 64.00 | **20.00** | **21.00** | 100.00* |

# 4 More Visual Results.

Fig.2, 3, 4 show more visual results of our GreedyFool on both ImageNet and CIFAR10 dataset with $\epsilon = 255, 10$ respectively. Besides the adversarial sample, we also show its corresponding distortion map $\varrho$, perturbation weight map $\mathbf{p}$, and adversarial noise.

As we illustrated in Sec3.2, the distortion map only concerns the invisibility of the image, so it is relatively sharp that only a few pixels have a large value. If we use it as the guidance directly, the perturbed pixels are limited to these small regions which have a larger value. This will influence the sparsity greatly. To make a trade-off between invisibility and sparsity, we calculate the perturbation weight map with the distortion map by Eq.3 in the main paper. Compared with the distortion map, we find the perturbation weight map is much smoother. This gives more choice of the pixels for attacking and improves the sparsity.

## Footnotes

*Work done during an internship at Microsoft Research Asia.

†Dongdong Chen is the corresponding author.

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

Figure 1: Input images and their corresponding distortion map, here we also show its $\sigma$-map used in $\text{PGD}_0$ [2].

| Original image | Distortion map | Perturbation weight map | Adversarial sample | Adversarial noise |
|---|---|---|---|---|

Figure 2: More visual results of our GreedyFool with $\epsilon = 255$ on ImageNet dataset. Besides the adversarial sample, we also show its corresponding distortion map $\varrho$, perturbation weight map $\mathbf{p}$, and adversarial noise(We enlarge the noise size to $2\times2$ pixels for more visible.)

Figure 3: More visual results of our GreedyFool with $\epsilon = 10$ on ImageNet dataset. Besides the adversarial sample, we also show its corresponding distortion map $\varrho$, perturbation weight map $\mathbf{p}$, and adversarial noise.

ε=255                    ε=10

Figure 4: More visual results of our GreedyFool with $\epsilon = 10, 255$ on CIFAR10 dataset. Besides the adversarial sample, we also show its corresponding distortion map $\varrho$, perturbation weight map $\mathbf{p}$, and adversarial noise.