[Reviews · NeurIPS 2020]

Review 1

Summary and Contributions: The paper proposes a new white-box adversarial attack, GreedyFool, to produce sparse perturbations. It minimizes the number of pixels perturbed and uses a GAN, which learns where the original image can be modified, to improve the invisibility of the attacks.

Strengths: - The idea of using a GAN to learn the distortion map and improve the invisibility of the attacks is novel and valuable. - In the experiments, GreedyFool seems to improve upon the competitors in both sparsity and invisibility of the attacks.

Weaknesses: - The method requires to train a GAN on the same dataset on which the attack is tested (while the \sigma-map used by the closest existing methods \sigma-PGD_0 is based only on the target image), requiring additional computational cost and knowledge of the training set. - There are some concerns about how the competitors are used.

Correctness: The methods and claims seem correct and mostly supported by the experiments.

Clarity: The paper is well written and easy to follow.

Relation to Prior Work: The paper discusses previous works, highlighting the novelty of the proposed approach.

Reproducibility: Yes

Additional Feedback: - The authors write that the gradient step which uses both direction and magnitude of the gradient is different compared to the commonly used sign of the gradient. However, the sign of the gradient is typical of Linf-attacks, while e.g. for L2-attacks the L2-normalized gradient is the standard choice. Moreover, also PGD_0 uses the gradient normalized wrt the L1-norm, which is similar to Eq. (5) assuming that the absolute value indicates the L1-norm (this should be clarified). - In the experiments, different values \epsilon are used as the maximum elementwise change, so that in practice the threat model consists of Linf + L0 constraints. Then, I think that in this case a more proper comparison would be with PGD wrt L0+Linf also introduced in [11]. Conversely, \sigma-PGD_0 allows larger changes in areas with are supposed to be invisible and has a different hyperparameter \kappa to tune the magnitude of the changes. Could the authors comment about this? - The results of PGD_0 in Table 2 seem worse than what indicated in [11] (the same NiN architecture is used), e.g. at m=10 pixels perturbed the success rate in [11] seems above 90%. Could the authors clarify this discrepancy? Also, on CIFAR-10 [11] report that the best results are achieved by CornerSearch (which also minimizes the number of perturbed pixels). Shouldn't this be included in the comparison? - Did the authors tried also to compare the proposed method to \sigma-CornerSearch for invisible attacks? - In many cases there is a missing blank space before brackets e.g. L51, L59. Overall, I think the paper is clear and the method rather effective. The experimental and theoretical comparison to previous works should be more clear, which would strengthen the paper. ### Update post rebuttal ### I thank the authors for answering to all my questions. Here some detailed comments. - GAN for distortion map, results of PGD_0: I'm satisfied with the replies. - Analysis sign vs direction: I still think that the sign of the gradients is specific to the Linf case (so I'd expect it to perform poorly for other norms), but I understand that the authors want to support the choice of using the direction with an ablation study. - Results of PGD wrt L0+Linf: although partial, they seem to still favor the proposed method, strengthening the claims of the paper. - Usage of \sigma-PGD_0: the authors seem to assume that an Linf-constraint is necessary for invisible perturbations, while the idea behind \sigma-PGD_0 is to allow change according to \sigma-map, without hard constraints. Also, even if one wants to add a pixelwise bound of \epsilon, setting \kappa=\epsilon, as done according to the rebuttal, doesn't seem the right way to achieve it. In fact, the maximal perturbation would then be smaller than \kappa, as it is multiplied by \sigma which is computed as the square root of the standard deviation of 3 values in [0, 1]. This seems confirmed also by the Linf-norm reported in Table 5 for \sigma-PGD_0, which is less than half of the given budget of 255, differently from all the other attacks. - CornerSearch: it achieves better results than the proposed method on CIFAR-10, hence I think it should be mentioned in the comparison regardless of being black-box or the runtime (which is anyway reasonable), together with \sigma-CornerSearch. I think that the paper has merit and the method is reasonable, effective and with a good degree of novelty. Also, I appreciate the thorough comparison of the imperceptibility of the attacks. However, the rebuttal, while clarifies some aspects, leads to other questions about the experimental comparison to other methods, which definitely needs to be clarified. The impression I have is still positive despite the issues mentioned above, but I cannot increase the score.


Review 2

Summary and Contributions: The goal of the work is to create sparse adversarial examples that only perturb a subset of pixels in an image. This is addressed in a two stage approach. Stage 1 chooses pixels to modify based on gradient information of the model and Stage 2 greedily drops low importance pixels to achieve greater sparsity and imperceptibility. More specifically, the method finds a (sparse) set of sensitive candidate pixels to perturb using gradient information from the model w.r.t. the input, then uses masks and a GAN to ensure that the perturbation constructed from this set of sensitive pixels is imperceptible and adversarially successful. The results are that GreedyFool can achieve higher attack success across different perturbation budgets than competing methods. It is also not restricted to perturbing only a fixed number of pixels and can work as a targeted and non-targeted attack.

Strengths: - I think the novelty and method are a strength of the work. Specifically, training a GAN to generate a distortion map to minimize the perceptibility of the noise. Also, GreedyFool can be used as a targeted and non-targeted attack whereas SparseFool is only untargeted. - In the experimental results, GreedyFool compares favorably to competing sparsity-based adversarial methods. It is also a strength that the method can scale to ImageNet and is not just restricted to CIFAR10. - In addition to just attack success rates, there is also consideration for quantitatively measuring perceptibility, which is interesting.

Weaknesses: - The main weakness is that the explanation of the methodology is confusing (Section 2), which I believe has to do with the order of how things are explained (e.g., algorithm before notation), the lack of top level diagram(s) which incorporate notation from the writing, and the sheer quantity of symbols and notation used is overwhelming. Specifically, without diagram(s), the interplay between g_t, m, \varrho, p, R is unnecessarily confusing. The method seems more complex than it actually is as a result of the writing. - Inadequate description of related works - It is unclear if the results change significantly when different pretrained networks to attack are considered? Results for only 1 network per dataset are shown, and although there is a citation for it, the NIN network seems somewhat atypical to use as the one and only model to evaluate CIFAR10 results on. What about very common networks like RN50, VGG16, DenseNet?

Correctness: Yes, I believe the claims about attack success are correct and the empirical methodology seems reasonable.

Clarity: No, I believe clarity is the main issue with this work. As mentioned, the methodology section is quite confusing with the introduction of many symbols and notations and lack of diagrams tying the notation together. The lack of clarity would also make the reproduction of this work difficult (but still possible). This paper would benefit greatly from a serious overhaul of the methodology section and I believe would then be quite good.

Relation to Prior Work: Not as clear as one may like: there is some discussion about how the work compares to related methods but it is scattered throughout the paper as opposed to being organized in a related work section (actually, there is no related work section).

Reproducibility: Yes

Additional Feedback: - put a space in-front of all (parenthesis) - remove trailing "." from some section headers - some wording and grammar issues (e.g., line 109) - in the future, it would be interesting to see if this noise transfers in a blackbox setting ******************************************************************************************** EDIT after rebuttal ******************************************************************************************** I think the author's responses regarding: (1) re-ordering some logical points in the methodology (for improved clarity), (2) attacking other ImageNet models, (3) testing blackbox transferability, and (4) using a general GAN-ImageNet for the distortion map, move this work in an overall positive direction. However, detailed discussion/comparison to related works may still be lacking, and I hope can be clarified in the final version. With this, I move my score to a "6: marginally above".


Review 3

Summary and Contributions: This paper proposes greedyFool to generate sparse pixel based adversarial examples by greedily selecting the positions to modify and dropping less important ones. The empirical results show that the attack is successful.

Strengths: The empirical results show the attack is more effective than the stoa attack.

Weaknesses: The contribution and novelty of the paper is quite limited. For instance, the proposed method aims to select the vulnerable positions in a greedy way which follows the same principle of deepfool, and the GAN based distortion map architecture is the almost the same with generating adversarial examples via GAN [1,2]. [1] Xiao, Chaowei, et al. "Generating adversarial examples with adversarial networks." IJCAI (2018). [2] Kos, Jernej, Ian Fischer, and Dawn Song. "Adversarial examples for generative models." 2018 ieee security and privacy workshops (spw). IEEE, 2018.

Correctness: yes

Clarity: yes

Relation to Prior Work: yes

Reproducibility: Yes

Additional Feedback: It would be interesting to compare with the state of the art one pixel attack [1], and it would be important to evaluate the transferability of the generated attacks to assess their blackbox attack ability. [1] Su, Jiawei, Danilo Vasconcellos Vargas, and Kouichi Sakurai. "One pixel attack for fooling deep neural networks." IEEE Transactions on Evolutionary Computation 23.5 (2019): 828-841.


Review 4

Summary and Contributions: This paper focuses on a sparse adversarial attack -- fooling deep neural networks by perturbing a few pixels of the input image. The authors propose a two-stage greedy-based method to select the positions to edit. The experiment part illustrates the effectiveness of the proposed method.

Strengths: +1 This paper is well written and easy to read. +2 Both GreedyFool and GAN based distortion map is overall reasonable +3 The experimental results are good.

Weaknesses: -1 The ablation study is difficult to understand. In section 3.3, it seems the authors didn’t compare their proposed GAN based distortion map with other kinds of distortion maps. Therefore, the reader can only know the importance of a distortion map, but not sure the necessity of using a GAN based distortion map.

Correctness: claims and method correct; empirical methodology correct

Clarity: yes

Relation to Prior Work: yes

Reproducibility: Yes

Additional Feedback: post-rebuttal update: The author's rebuttal has well addressed my concerns. The authors are highly encouraged to strengthen the ablation and experimental comparison in the updated version. Overall, I think this is a reasonably good paper. Therefore I will keep my initial score.

[Author Response · NeurIPS 2020]

We would like to thank all reviewers for their valuable comments. We appreciate that the novelty and effectiveness of
our method are well recognized by R2, R3, R5. All the minor grammar errors will be fixed in the final version.
**R2:*Dataset specific GAN for distortion map*** : Thanks, really good question. Inspired by it, we find that it is unnecessary
to train GAN for each dataset. A GAN trained on ImageNet (GAN-ImageNet) also works on other datasets. If we
generate the distortion map with GAN-ImageNet for CIFAR10 images by resizing, the mean perturbed pixel number is
6.01 and the median is still 4, almost the same as the result in our paper. We will add such analysis in the final version.
**R2:*$l_1$-norm problem***: Sorry, this is our negligence that we only consider the traditional $l_\infty$-norm method and SparseFool
in this part, we will improve it in the final version. But here we want to emphasize that to the best of our knowledge, we
are the first to quantitatively prove the importance of direction for sparse attack(in the ablation study part).
**R2:*Comparing with $PGD_0$ $l_0$+$l_\infty$***: In the main experiments, since we consider both sparsity and invisibility, we think
it is fairer to compare with the invisible version of the previous method (SparseFool with $\epsilon$ constrain and $PGD_0$ with
$\sigma$-map ). Here we report the result of $PGD_0$-$l_\infty$ with 200 pixels budget and $\epsilon = 10, 100$. Its fooling rate is 40.62% and
80.14% respectively, still lower than our GF. More results will be added to the final version.
**R2:*$\kappa$ for $\sigma$-$PGD_0$***: In Tab.1 and Tab.2, we already set $\kappa = \epsilon$ to control the max value of the perturbation for $\sigma$-$PGD_0$.
**R2:*Why $PGD_0$ performs worse than it in [11]***: All the results are obtained by running the official code released by
[11]. There are three key possible reasons that lead to the difference: 1) Data difference, different from traditional dense
attack, sparse attack performance is more sensitive to the input (some images only need perturb 1 pixel while some
need 100 pixels). [11] only uses 1000 test images, while we test the whole CIFAR10 test set. We asked for the advice
from the author of [11] and we both think this possibly leads to the performance difference. 2) The metric difference,
we found that [11] treat clean images classified incorrectly by the model as the successful attack samples. But we
ignore these images when computing the fooling rate (it will be a bit lower for the same results), this also influences the
result slightly. 3). Parameters difference, as there are some parameters of $PGD_0$ not reported in [11], we use the default
parameters of its official code, this may also influence the result.
**R2:*Sparsity comparison with CornerSearch[11]***: CornerSearch(CS) is a black-box method by traversing all the pixels,
so it is very slow. In contrast, since our method is a white-box method, there does not exist a proper or fair comparison
for success rate and speed. For CIFAR10 with $\epsilon = 255$, the mean pixel and time used by CS are 3.49 and 6.29sec, while
our GF is 5.98 and 0.114sec. CS performs better than our GF a bit but significantly slower. For ImageNet with $\epsilon = 255$,
the mean pixel and time used by CS are 104.3 and 644.95sec, but our GF is only 62.13 and 5.25sec.
**R2:*Invisibility Comparison with $\sigma$-CornerSearch[11]***: For invisibility, we need 12000 ImageNet images for the
invisibility evaluation, due to the short rebuttal time and $\sigma$-CS is relatively slow, we failed to generate enough samples.
We will add this result to the final version.
**R3:*Confusing notations***: Thanks for your suggestion, we will reorganize the logic of the method part and add more
necessary diagrams in the final version. The $g_t$ is the gradient of $x_t^{adv}$ at the $t$ iteration. $\mathbf{m}$ is a binary mask denoting
whether a pixel is selected or not. $\varrho$ is the distortion map generated by GAN and $\mathbf{p}$ is calculated from $\varrho$ for the balance
between invisibility and sparsity, visual results about $\varrho$ and $\mathbf{p}$ are shown in the supplementary materials.
**R3:*Lack of related work***: Sparse adversarial attack is a new direction and related works are relatively few, so we only
introduced them in the Introduction part due to the page limit. More related works will be added in the final version.
**R3:*Attack other models***: On ImageNet($\epsilon = 255$), our GF fool VGG16, ResNet50 and DenseNet161 with 21.98, 25.83,
34.69 pixels on average respectively. For SF, it needs 85.40, 103.61, 137.94 pixels respectively, nearly 4$\times$ than us.
**R3,R4:*Black-box transferability***: On ImageNet($\epsilon = 255$), we transfer from DenseNet161 to Vgg16 and ResNet50. For
SparseFool[26], its fooling rate is 26.76% and 15.38%, while our GF is significantly better with 40.33% and 30.67%.
**R4:*Similar to DeepFool and GAN-based attack method [1,2]?***: *No, we cannot agree on the comment*. Our method
is totally different from DeepFool and the cited methods from the hypothesis and task perspective: 1) DeepFool's
hypothesis is that, if the input is perturbed with the direction toward the nearest hyperplane in each iteration, the
classifier can be fooled with the smallest dense perturbation under $l_{1,2}$-norm. *But for our GreedyFool, the hypothesis is*
*that the pixel that has largest value of the gradient in each iteration influences the prediction most, if we perturb it,*
*we can fool the classifier with the smallest perturbation pixel number under $l_0$-norm.* 2) The task of [1] is generating
adversarial samples by GAN, and the task of [2] is how to generate adversarial samples to fool the generative model. In
contrast, for our GAN-based distortion map part, *it instead acts as the guidance to select the pixel with the minimal*
*modification visibility and we train it by adding a global perturbation noise to it in an adversarial way.*
**R4:*Comparison to One Pixel-Attack***: One Pixel Attack(1-PA) is a black box attack with high computation cost and
needs predefined pixels budget. To attack ResNet18 on ImageNet with 50 pixels budget, the success rate of 1-PA is
60.87% with 120sec per image. While the success rate of our GF is 88.75% with only 0.6sec per image.
**R5:*Comparison to other distortion map***: We have already shown the visual result and corresponding analysis in the
supplementary materials. Here we further report the quantitative result by replacing the GAN-based distortion map
with the $\sigma$-map and generate adversarial samples with $\epsilon = 10$. The median perturbation number and SRM detection
rate of the $\sigma$-GF is 301.50 and 57.20%, which is worse than our GF which is 222.50 and 54.00% respectively.

[Meta-Review · NeurIPS 2020]

This paper proposes a method to generate L0 adversarial examples. The method is efficient and better than simple baseline attacks, but the evaluation is not thorough with respect to prior l0 attacks, and artificially weakens the comparison to PGD/CW by restricting the number of iterations. However, the reviewers by and large liked the paper and found the method useful. The rebuttal addressed many of the concerns and the updated paper likely will better reflect the novel contributions of the defense. The fact that this method is efficient, while still performing roughly on par with optimization attacks, indicates better optimization L0 attacks should be possible as well.